# ELASTIC LOAD BALANCING FOR DYNAMIC LLMS

## ABSTRACT

To reduce the computational and memory costs of Large Language Models (LLMs), families of training schemes that introduce dynamic training workloads is emerging. For example, in gradual pruning, the pruning of the parameters of a model happens during training to reduce resource requirements. However, one of the side effects of this is that sparsification introduces workload imbalance among workers, which, in turn affects the pipeline parallelism efficiency in distributed training. Similar issues arise in layer freezing schemes. We propose load balancing algorithms to adaptively maintain equal compute workloads among different workers, and also dynamically pack work into fewer workers while sustaining training throughput. Our solution, DYNPIPE, supports both single nodes with multi-GPUs and also systems with multi-nodes. Our methods accelerate the training of dynamic GPT class of models by up to 1.29x in a single node with 8 A100 GPUs, and 2.54x in a data and pipeline hybrid parallelism multi-node setting up to 720 A100 GPUs, over state-of-the art production solutions used in training static LLMs. DYNPIPE is available at `https://anonymous.4open.science/r/DynPipe-CC54`

## 1 INTRODUCTION

Sizes of neural networks used to train LLMs has exponentially grown since the first attention-based model Vaswani et al. (2017). This growth demands more memory and compute power. Yet, neither the memory capacity nor the compute capability of a single accelerator increases at the same rate Sevilla et al. (2022). As a result, high-performance computing centers and cloud providers use a mix of model and data parallelism for training large models Narayanan et al. (2021). One of the most commonly used forms of model parallelism in language models is pipeline parallelism, in which consecutive layers are grouped into stages, with each stage assigned to one accelerator (worker) Kahira et al. (2021). Input mini-batches are split into micro batches (chunks) to improve accelerator utilization by overlapping computation in a pipeline fashion Huang et al. (2019); Harlap et al. (2018); Fan et al. (2021); Li & Hoefler (2021).

In traditional LLMs training schemes, the workload for each pipeline stage is known in advance and remains static throughout the training. To reduce computational resource requirements, new training schemes that introduce dynamic training workloads are emerging. This includes: **a)** gradual pruning where the parameters of a model are pruned (i.e. sparsified) during training Gale et al. (2019), **b)** freeze training where some of the layers of the model are adaptively frozen during training Wang et al. (2022), **c)** neural networks where different input samples take different pathways through the model layers, e.g. gated neural networks Shazeer et al. (2017), sparsely activated Mixture of Experts (MoEs) Zhou et al. (2022b), Switch Transformers Fedus et al. (2022) etc. Other than computational efficiency, there is a wide range of reasons that motivate the use of different forms of dynamic models to improve certain model attributes, such as explainability and generalization. We refer the reader to the survey by Han et al. (2021) on different forms of dynamic models.

One of the unforeseen side effects of these dynamic models is that they introduce load imbalance in pipeline parallelism, effectively decreasing the throughput of LLM training Zhou et al. (2022a); He et al. (2022). For example, Figure 1 (left) shows the maximum idleness of GPUs for GPT language models with different numbers of layers, where the models are sparsified up to 90%. Load imbalance increases as the model is pruned further, which reduces pipeline utilization. Load imbalance can manifest itself as bubbles that appear in the pipeline due to a stalling accelerator waiting to receive work from its late neighboring worker. At 90% sparsification, the idleness ratio of a pipeline on a

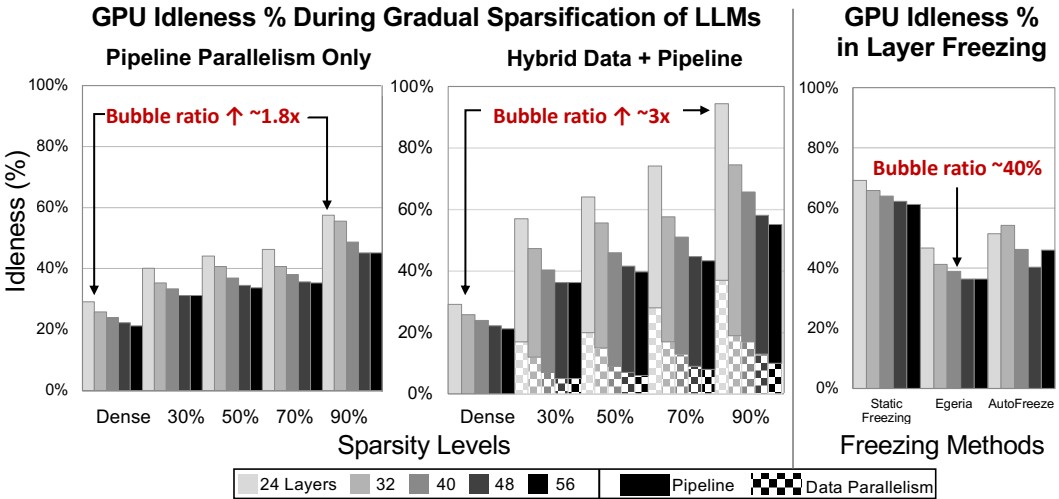

Figure 1: Idleness percentage of GPUs for a single training iteration of GPT models Radford et al. (2018) parameterized to have between 24 and 56 layers with two example cases of dynamicity: gradual sparsification and layer freezing. **Left**: Pipeline parallelism on 8 A100 GPUs using the highest performing pipeline parallelism scheme known to the authors (Chimera) Li & Hoefler (2021). Due to load imbalance, we observe almost a two fold increase in idleness at 90% sparsity levels. Note that idleness at *Dense* is the inherent pipeline bubbles of a static model. **Middle**: Idleness percentage in hybrid parallelism using 720 GPUs (x8 A100 GPUs × 90 nodes): 90-way data + 8-way pipeline parallelism. Imbalanced pipelines in dynamic models lead to additional stalling in data parallelism for the allreduce collective used in the gradient exchange after each iteration. **Right**: Idleness percentage in layer freezing on 8 A100 GPUs. State-of-art layer freezing schemes that incorporate load balancing (Egeria( Wang et al. (2022) and AutoFreeze Liu et al. (2021)) yield ∼40% bubble ratio.

single node with 8 A100 GPUs can be as high as 57%, as shown in Figure 1 (left). Since a pipeline is only as fast as its slowest stage, load balancing becomes crucial for resource utilization.

To make matters worse, if a pipeline and data parallelism hybrid scheme is used, as is typical in production-level training of LLMs Narayanan et al. (2021), the bubbles in the unbalanced pipeline would further penalize the end-to-end training time since they introduce irregular stalling to the allreduce collective used in data parallelism to average the gradients. Figure 1 (middle) shows a bubble ratio of more than 90% in hybrid parallelism. In another example involving dynamic models, Figure 1 (right) illustrates bubble ratios reaching 40% due to the load imbalance attributed to dynamic layer freezing, even when state-of-the-art solutions for load balancing models with frozen layers, such as Egeria Wang et al. (2022) and AutoFreeze Liu et al. (2021), are employed.

State-of-the-art production frameworks typically implement a static load balance at the beginning of training and maintain the same load distribution throughout the training. For instance, Megatron-LM Shoeybi et al. (2019) evenly distributes all transformer layers across the accelerators. Deep-Speed Smith (2023) currently offers three partitioning methods for distributing model layers: *Uniform*, which balances the number of layers; *param*, which balances the number of parameters in each stage; and *regex*, which balances layers whose names match a given regex pattern. However, this approach operates on the assumption that the accelerators' workloads remain roughly consistent throughout training. As a result, it fails to address the pipeline stalls introduced by dynamic models, ultimately leading to a decrease in computational efficiency.

Considering the increasing importance of efficient sparse dynamic models, layer freezing, and other dynamic training workloads, this work aims to mitigate the pipeline stalls introduced by dynamic models. We introduce DYNPIPE, an elastic load-balancing framework designed for dynamic models, to ensure balanced pipeline stages during training. DYNPIPE dynamically redistributes the workload among accelerators whenever an imbalance arises during training, consequently enhancing computational efficiency and leading to cost savings. DYNPIPE incorporates two different dynamic balancers, both proven to converge to the optimal workload balance among workers. Our experi-

ments demonstrate that DYNPIPE incurs negligible overhead and can scale effectively in both: a) single-node multi-GPU environments and b) multi-node multi-GPU environments typically used for training LLMs with hybrid parallelism.

DYNPIPE not only enhances performance through dynamic load balancing but also offers the capability to elastically adapt GPU resources. Specifically, as the total workload decreases during training due to sparsification, the load balancer consolidates the work onto fewer GPUs –subject to memory capacity constraints– while maintaining performance. GPUs that are no longer needed for training can then potentially be released back to the job scheduler. For example, in single-node multi-GPU systems, Nvidia Multi-Instance GPU (MIG) Nvidia (2023) supports node partitioning for multi-tenancy. GPUs that have been released can be returned to MIG for allocation to other tenants. In multi-node environments, cloud schedulers have the ability to acquire released resources and reassign them to other jobs, often leveraging technologies like elastic Kubernetes Elastic (2023).

**DYNPIPE offers a solution that empowers researchers to explore dynamic models and it is the first work to study pipeline stalls caused by unstructured sparsity during dynamic training. Research on dynamic models will not deliver practical impact unless there is a platform from which those models can be made efficient.** DYNPIPE provides the essential platform for achieving efficiency in these models, offering a unique opportunity to support innovative ideas in dynamic pruning and beyond. Additionally, considering the substantial costs, often tens of millions of dollars, required for each training run of GPT-class models Li (2022); Heim (2022); Morgan (2022), improving the efficiency of dynamic models can result in significant cost savings.

Finally, we emphasize that DYNPIPE functions as a solution complementing pipelining, pruning, and layer freezing schemes. It has no impact on model accuracy, as its role is solely to redistribute workload without interfering with the learning process. DYNPIPE's load balancing method operates independently of the pruning or freezing approach, making it compatible with various dynamic schemes. It can even be applied to models that undergo dynamic changes for reasons other than sparsification, such as manufacturing variability of computing units Sinha et al. (2022), or those that involve sparsely activated mixtures of experts Zhou et al. (2022b). In short, our contributions are:

- We introduce DYNPIPE, which enables researchers to explore dynamic models and significantly improves the end-to-end training efficiency of such models, making their practical application more feasible. DYNPIPE is orthogonal to the underlying pipeline parallelism, pruning, freezing schemes; allowing for compatibility with various compute optimization/reduction schemes.

- We propose load balancing algorithms proven to converge to optimal balancing in order to alleviate the negative effects of dynamic models on pipeline utilization.

- We show the benefits of the framework with a gradual pruning training and layer freezing scenarios in both single-node and multi-node settings. We further introduce a scheme for reducing the number of GPUs used during training by re-packing work to fewer GPUs.

- DYNPIPE achieves close to 1.3x speedup over Megatron-LM on a single-node with 8 A100 GPUs, and more than 2.5x speedup for multi-node hybrid data and pipeline parallelism with up to 720 A100 GPUs. Additionally, the framework achieves an average speedup of 2.3x over the state-of-the-art layer freezing solution. We demonstrate that the re-packing strategy is proven effective in reducing the number of GPUs by half while sustaining comparable performance.

## 2 MOTIVATION AND BACKGROUND

### 2.1 BUBBLES IN PIPELINE PARALLELISM

There are two types of bubbles in pipeline parallelism: (i) inherent bubbles of the pipeline schedule (e.g. bubbles in-between forward and backward passes in GPipe Huang et al. (2019)), and (ii) bubbles introduced by the dynamic models during training (e.g. bubbles introduced by sparsification during training). We aim to reduce the latter type of bubbles by carefully redistributing the layers among stages to minimize the workload imbalance in the pipeline. Appendix A elaborates with analysis of bubbles in pipeline parallelism.

## 2.2 DYNAMIC MODELS

To reduce computational and memory costs, training schemes that introduce dynamic training workloads have started to emerge. One of the dynamic training schemes is gradual pruning to reduce the model size. In a gradual pruning scheme, the number of parameters used changes during training based on a pruning strategy. If this pruning technique does not prune each layer uniformly (e.g. global magnitude pruning Hagiwara (1993)), the workload of each stage may be significantly different, which may introduce extra bubbles (stalls) in the pipeline Zhu & Gupta (2017); Frankle & Carbin (2018); Bellec et al. (2017).

Another emerging dynamic training scheme is freeze training which relies on the idea that some layers of a network might converge faster than others, and hence can be frozen and excluded from the model during training Shen et al. (2020). If the frozen layers are not evenly distributed among accelerators, this can act as a source of imbalance in the pipeline as reported by Shen et al. (2020).

## 2.3 NEURAL NETWORK PRUNING

The lottery ticket hypothesis states that there exists sub-networks with dense over-parameterized networks that can be trained to the same accuracy Frankle & Carbin (2018); Gale et al. (2019). Network pruning is a sparsification procedure that removes a fraction of the parameters to achieve the same performance with a smaller network. Pruning during training leads to dynamic models. Appendix B elaborates on the factors that drive pruning: pruning criteria, structure, schedule.

## 2.4 LAYER FREEZING

The eailer layers of Deep Neural Networks (DNNs) tend to train faster than later layers Wang et al. (2022). This phenomenon arises from the progression of DNN features, which shift from a generic understanding to a task-specific one, spanning from the initial to the final layer Yosinski et al. (2014). Consequently, the front layers of a DNN frequently reach convergence swiftly, whereas the deeper layers necessitate a substantially more extended training period, a characteristic commonly noted in both vision and language models [75, 79]. By freezing the state of these early-converged DNN layers, the computational cost can be reduced without compromising model accuracy Liu et al. (2021); Wang et al. (2022).

# 3 DYNPIPE: ELASTIC LOAD BALANCING FOR DYNAMIC LLMS

## 3.1 OVERVIEW

In this work, we take pipeline parallelism with gradual pruning (sparsification during training) and layer freezing as two example cases of dynamic models, for which current execution systems in DNN training are not ready to handle efficiently. Even though we show the efficiency of our load balancing system for dynamic DNNs with these example cases, they can be a basis for expanding to other forms of dynamic models, such as MoEs.

Algorithm 1 shows the overall flow of operations of DYNPIPE with gradual pruning. The algorithm takes as input a model, the number of training iterations, the rank of the accelerator, and several arguments for pruning, balancing, and packing the model's workloads. We start the training with the original model and train it until a user-specified pruning region (an iteration range e.g. 3000-7000) is reached (line 7-8). The model is pruned only if the training is in this pruning region. Once the training is in the pruning region, the model parameters are gradually pruned every $prune\_freq$ iteration (e.g. every 1000 iterations) where $prune\_freq$ is the frequency of pruning until the sparsity of the model reaches the given target sparsity (line 9-16). The first iteration after each pruning operation is used for profiling the time it takes to execute each layer in the pruned model and the memory usage of all accelerators in the pipeline. Next, DYNPIPE collects the profiling information and decides on balancing the workload by moving layers across pipeline stages based on the execution times of individual layers to minimize the pipeline stalls, subject to the constraints of memory capacity per worker (line 17-20). DYNPIPE also attempts to re-pack the total workload into fewer number of GPUs if the re-packing feature is enabled by the end user (line 21-23). Once the training is out of the pruning region, the balanced pipeline continues to execute with the pruned model.

---

**Algorithm 1** End-to-end Training of Dynamic LLMs with DYNPIPE

---

    **Input:** model, train_iters, rank
    **Input:** prune_args, balance_args, pack_args
 1: prune_rat, prune_region, prune_freq ← prune_args
 2: is_load_balance, balancer ← balance_args
 3: is_pack, num_gpus_to_pack ← pack_args
 4: prune_idx ← 0
 5: prune_iter ← NULL
 6: profile ← 0
 7: **for** iter ← 0 **to** train_iters **do**
 8:    train_step(model, profile)
 9:    **if** iter **in** prune_region AND iter % prune_freq == 0 **then**
10:       g_prune(model, prune_rat[prune_idx], rank)           ▷ Algo. 1 in Appendix B
11:       prune_idx += 1
12:       prune_iter = iter
13:       **if** is_load_balance **then**
14:           profile = 1
15:       **end if**
16:    **end if**
17:    **if** is_load_balance AND iter == prune_iter + 1 **then**
18:       load_balance(model, balancer)                ▷ Algo. 2 in Appendix B
19:       profile ← 0
20:    **end if**
21:    **if** is_pack AND iter == prune_iter + 1 **then**
22:       pack_workload(model, num_gpus_to_pack)       ▷ Algo. 3 in Appendix B
23:    **end if**
24: **end for**

---

Figure 2 illustrates the overview of DYNPIPE with all its steps. The implementation of individual steps of global pruning, layer freezing, load balancing, and re-packing can be found in their respective sections.

## 3.2 GRADUAL GLOBAL MAGNITUDE PRUNING

For our pruning design, we use the gradual pruning schedule proposed in Zhu & Gupta (2017) which is formulated as:

$$S_t = S_f + (S_i - S_f)(1 - \frac{t - t_0}{n\Delta t})^3, \quad t \in \{t_0, t_0 + \Delta t, ..., t + n\Delta t\} \tag{1}$$

To our knowledge, there is no deep learning framework that supports global pruning on a distributed model at the time of this writing (support is only for undistributed models). Appendix B elaborates on the unstructured magnitude pruning scheme we implemented in PyTorch.

## 3.3 LAYER FREEZING

DYNPIPE sits on top of layer freezing solutions. More specifically, we build on Wang et al. Wang et al. (2022) Egeria solution by monitoring the rate by which the training loss changes, freezing layers when they reach the convergence criterion, and drop frozen layers from in both the back propagation phase and gradient exchange when data parallelism is used. It is important to note that Egeria periodically updates the reference model (on the CPU) to drive the layer freezing, yet does not actively try to remedy the load imbalance caused by layer freezing. The effect of load imbalance is particularly pronounced since earlier layers tend to be more frozen than later layers, i.e. the layer freezing is not uniformly occurring across the model. In comparison, DYNPIPE load balances dynamically, and in an orthogonal fashion, the spread of layers on GPUs every time the reference model that drives the freezing is updated.

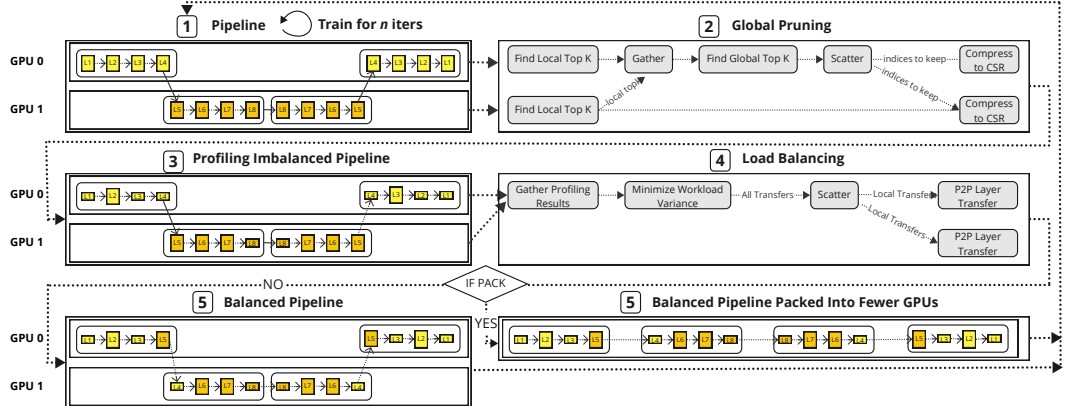

Figure 2: Overview of DYNPIPE. The flow in the figure (top to bottom) is repeated until the target sparsity is reached or training is completed. Each yellow and orange rectangle represents a transformer layer (i.e. encoder or decoder layer). The size of a rectangle illustrates the amount of work in a layer. (1) shows the pipeline before pruning and trains the model for $n$ iterations (2) performs global gradual pruning, (3) profiles the pipeline to check if there is any imbalance between stages, (4) performs load balancing based on the profiling results, (5) trains the balanced pipeline until the next pruning, optionally it reduces the number of resources (GPUs) used in training by re-packing.

## 3.4   LOAD BALANCING

DYNPIPE implements two load balancing algorithms, and can be extended to support other algorithms. The first is centralized parameter-based partitioning that balances partitions based on the number of parameters. We also implemented a variant where the same algorithm balances partitions based on the layer execution times, instead of the number of parameters. The two variants of this algorithm are built on top of DeepSpeed's load balancing utility functions for partitioning in model parallelism Smith (2023). The second algorithm is an iterative decentralized diffusion-based algorithm that aims to minimize the variance between the workload of each rank by attempting to move layers from overloaded GPUs to underloaded ones in an iterative way. The workload can either be the layer execution times or the parameter counts as in the centralized partitioning method.

We demonstrate that the two load balancing schemes (used in Algorithm 1) meet the goals for optimal load balancing by using the following lemmas. lemmas proofs presented in Appendix B.

**Lemma 1** *A centralized load balancer $L_c$ over $N$ workers satisfies maximum reduction in the imbalance $N_i$ if and only if $N_i$ reduces the bubble ratio to minimum.*

**Lemma 2** *An iterative decentralized diffusion based load balancer $L_d$ over $N$ workers satisfies maximum reduction in the imbalance $N_i$ if and only if $N_i$ reduces the bubble ratio to minimum. Also the load balancer is guaranteed to converge to the maximum reduction in imbalance in the following number of rounds*

$$O\left(\min\left\{N^2\log\left(\frac{SN}{\gamma}\right)\log N, \frac{SN\log N}{\gamma}\right\}\right)$$

*where $\gamma \in \mathbb{R}_{>0}$ is the convergence factor and $S \in \mathbb{R}_{>0}$ is the total number of stages in the pipeline.*

## 3.5   RE-PACKING DYNAMIC MODELS TO FEWER WORKERS

Workload re-packing is the process of merging the total workload into a smaller number of workers (GPUs) with the purpose of using the available resources more efficiently, i.e. unused resources can be released. This can be achieved with simple algorithms (in small scale) such as first-fit, best-fit, and round-robin as well as complex optimization heuristics. Workload re-packing aims to increase GPU utilization and reduce the overall number of GPUs employed to continue the training process. For long training schedules that are common in LLM training, workload packing can result in substantial

Figure 3: Throughput (tokens/sec) of end-to-end training while pruning of GPT models. Speedup is over the highest among static Megatron-LM and DeepSpeed. Comparison of different load balancing methods where the target sparsity is 90% in a gradual pruning setting. **Left (single-node)**: Using 8 A100 GPUs. Time-based dynamic load balancers outperform the baseline static load balancers and dynamic parameter-based load balancers in all model sizes. **Right (multi-node)**: A hybrid of data and model parallelism is used on 720 GPUs across 90 nodes: inter-node is data parallel and intra-node is model parallel. Load imbalance in static solutions leads to degradation in performance due to the effect of the imbalance on the gradient exchange collective in data parallelism.

cost savings. It may also provide improved performance due to reduction in the number of cross-GPU communication calls, and smaller pipeline bubbles.

We use a first-fit algorithm for workload consolidation. The goal of this algorithm is to reduce the number of active GPUs (subject to memory capacity constraints). When the combined memory usage of every pair of GPUs is less than the memory capacity of a single GPU, we migrate the layers in order to free one of the GPUs. This repeats in an iterative fashion for every two pairs until no more GPUs can be eliminated. Appendix B elaborates on our algorithm for efficient re-packing.

## 4 EVALUATION

This section contains empirical results and analysis of DYNPIPE's effectiveness. Experiments were mainly conducted on a supercomputer at which each of the compute nodes contains two Intel Xeon Platinum 8360Y processors, and eight 40GiB NVIDIA A100 GPUs. The GPUs in the same node communicate with CPUs using PCIe Gen 4 x16 per GPU, and NVSwitch amongst the GPUs (NVLink3 x12). The compute nodes are connected by 4 Infiniband HDR (200 Gbps). We used CUDA 11.3, OpenMPI 4.0.5, and PyTorch 1.12 with NCCL 2.9.9 distributed backend.

We train the model on the Wikipedia dataset Foundation (2023). All models used for training have a sequence length of 512, a hidden size of 1024, 16 attention heads, and the models are trained with a micro-batch size of 2 and batch size of 64 for 10000 iterations, unless specified otherwise.

We conducted experiments using two dynamic load balancing algorithms, each with two different configurations. These algorithms were employed consistently in both the gradual pruning and layer freezing experiments. The first algorithm, referred to as *Partition by Param*, is based on a DeepSpeed Rajbhandari et al. (2020) API. It uses a combination of binary search and linear probing to determine the optimal partitioning based on the parameter counts of the encoder/decoder layers. Another variation of this algorithm, called *Partition by Time*, employs execution times of encoder/decoder layers as input. The second algorithm is a decentralized iterative diffusion-based load balancing approach, which iteratively minimizes load variances among accelerators. Similar to DeepSpeed, this balancer has two variants: *Diffusion by Param* and *Diffusion by Time*.

For the multi-node experiments, as we increase the number of GPUs, we also increase the batch size to fix the number of micro batches to four times the number of GPUs in the pipeline, as suggested in Huang et al. (2019) to achieve good pipeline utilization.

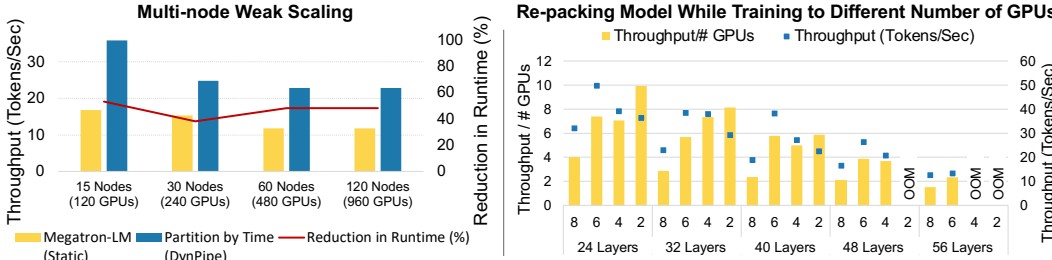

Figure 4: **Left**: Gradual pruning weak scaling throughput (tokens/sec) comparison of baseline static load balancing with Megatron-LM and dynamic load balancing with *Partition by Time* algorithm of DYNPIPE. Left y-axis: throughput. Right y-axis: speedup of *Partition by Time* over Megatron-LM. **Right**: Re-packing the layers of gradually pruned GPT models into fewer GPUs as the model gets smaller. Left Y-axis: throughput/number of GPUs. Right Y-axis: throughput (tokens/sec).

### 4.1 END-TO-END TRAINING WHILE PRUNING: SINGLE-NODE WITH MULTI-GPUS

We trained GPT models Radford et al. (2018) having different numbers of layers with eight A100 GPUs in a single node. The pruning region starts from iteration 3000 and continues until iteration 7000 and the model is pruned every 1000 iterations until the 90% target sparsity is reached. This corresponds to sparsity levels of 52%, 79%, and 90% after each pruning step. All other hyperparameters are the same as Megatron-LM.

Figure 3 (left) presents the throughput of two static and four dynamic load balancers. The first static balancer, Megatron-LM Shoeybi et al. (2019), evenly distributes layers across accelerators. The second static balancer, DeepSpeed Microsoft (2023), balances the number of parameters before training begins. In contrast, the dynamic balancers (*Partition by Time*, *Partition by Param*, *Diffusion by Time*, and *Diffusion by Param*) redistribute layers after each pruning step. Parameter-based balancers require profiling after the pruning step for memory usage information, while time-based balancers require profiling for memory usage and layer execution time information.

As depicted in the figure, the use of layer execution time for dynamic load balancing, such as diffusion or partitioning, consistently outperforms parameter count-based implementations across all scales. In every scale, execution time-based dynamic balancers surpass the baseline static balancers. However, parameter-based dynamic balancers occasionally exhibit slowdowns during training, as seen with *Partition by Param*. This behavior can be attributed to the fact that as transformer layers are pruned, parameters in the embedding layer of the first GPU and the post-processing layers of the last GPU become dominant in terms of parameter counts. This causes parameter count-based algorithms to overly redistribute the transformer layers of the first and last GPU to other GPUs, often more than necessary. **In summary, time-based load-balancing algorithms consistently achieve higher throughput in all cases and surpass the baseline static balancers.**

### 4.2 END-TO-END TRAINING WHILE PRUNING: MULTIPLE-NODES WITH MULTI-GPUS

Figure 3 (right) shows the training throughput for the multi-node hybrid parallelism case. As with the single-node case, using layer execution time for diffusion or partitioning dynamic load balancing outperforms the parameter count-based implementations in each scale, for up to 2.54x. The speedup over the static baseline is higher than the case of single-node due to the reduction in stalling exhibited by the data parallelism allreduce of gradients when the pipeline is balanced.

For multi-nodes with multi-GPUs weak scaling experiments, we trained the GPT models having different numbers of layers and batch sizes on up to 90 nodes each of which contains 8 A100 GPUs. The pruning region starts from iteration 30 and continues until iteration 70 and the model is pruned every 10 iterations until the 90% target sparsity is reached. The pruning and load balancing overheads are excluded from the measurements since the number of iterations to do this scaling experiment is not sufficient enough to amortize the overheads; in actual training (1000s to 10,000s iterations) the pruning and load balancing overheads would be negligible (elaborate overhead analysis in Appendix D). Figure 4 shows that the pipeline that is dynamically balanced with *Partition by*

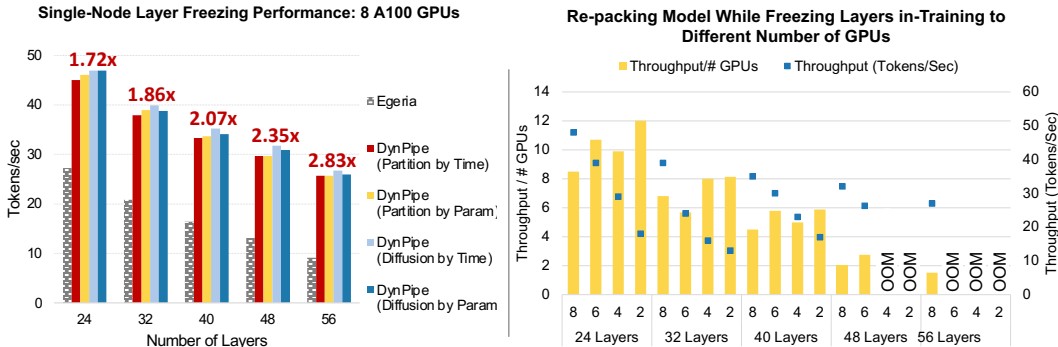

Figure 5: **Left**: throughput of end-to-end training while freezing layers of GPT models. Comparison of different balancing types for 8 A100 GPUs in a single node. **Right**: Re-packing the layers while freezing layers of GPT models into fewer GPUs as the model gets smaller when layers are frozen. Left Y-axis: throughput/number of GPUs. Right Y-axis: throughput (tokens/sec).

*Time* algorithm of DYNPIPE reaches higher throughputs in all scales and it provides speedups over baseline Megatron-LM up to 2.12x.

### 4.3 END-TO-END TRAINING WITH LAYER FREEZING: SINGLE-NODE WITH MULTI-GPUS

Figure 5 shows the speedup of using DYNPIPE over Egeria Wang et al. (2022), the state-of-art solution for layer freezing. We can observe two main things. First, different load balancing algorithms speedups over static algorithms are mostly the same, mostly due to the same load balancing solutions the different algorithms arrive at when entire layers are frozen. Second, the speedup reported by DYNPIPE means that more than half of the bubble appearing due to freezing layers can be eliminated, which empirically demonstrates the effectiveness of DYNPIPE.

### 4.4 RE-PACKING MODELS TO FEWER GPUS

In the re-packing experiments, the training starts with 8 GPUs and after each pruning step, DYNPIPE attempts to re-pack the total workload into fewer GPUs while satisfying the memory capacity constraints. Figure 4 (right) reports the throughput/number of GPUs for each model size where the model is packed into 6, 4, and 2 GPUs. The 8 GPU setting for each model size serves as a baseline where there is no re-packing. This measurement also corresponds to the performance per dollar metric as the cost is directly proportional to the number of GPUs used in training.

We observe that in all model scales (e.g. 24 or 32 layers), re-packing can allow the training to be continued with fewer GPUs which may result in significant cost savings. For example, in Figure 4, reducing the GPU count from 8 to 4 results in almost the same throughput while the resource usage cost is reduced by 50% for 32 layers case. The benefits of re-packing are not limited to the cost savings. For instance, re-packing from 8 to 6 GPUs in 24 layer setting also increases the throughput, which results in faster training time. **In other words, re-packing the workload into fewer GPUs after pruning may lead to faster or comparable training time with fewer resources**.

Figure 5 (right) shows the re-packing effect for layer freezing experiments. We can observe higher throughput/#GPUs since layers freezing is more regular than global pruning, i.e. re-packing is more efficient when entire layers are removed in comparison to pruning some of the layers parameters.

### 5 CONCLUSION

DYNPIPE is a load-balancing system for dynamic models where the loads of the workers change during training. DYNPIPE provides better load balance than state-of-the-art static load balancing approaches, which results in better efficiency and faster end-to-end training time. Empirical results for LLMs with a gradual pruning and layer freezing in-training show that DYNPIPE significantly improves the training throughput over the counterparts. We foresee that dynamic models will be more prominent in the future and dynamic load distribution will be of utmost importance.

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
