# Elastic Load Balancing for Dynamic LLMs
## Supplementary Material

## A  Bubble Ratio in the Static Model

In this Section we describe the theoretical bubble ratio that appears in the static model. The *bubble ratio* refers to the ratio of the idle time of devices when different workers (GPUs) stall while waiting for work to be available. Additional bubbles appear in the pipeline when the model become dynamic; due to pruning while training. The bubble ratio for the Chimera Li & Hoefler (2021) (2-model duplicate) pipeline scheme we use in the paper is:

$$\frac{\left(3\left(\frac{P^2}{2}-P\right)+6P-2\left(\frac{P^2}{2}-P\right)/P-8\right)T_C+2\left(\frac{P^2}{2}-P\right)T_B}{(3P^2-2P)T_C+(2P^2-2P)T_B+P^2T_F} \tag{1}$$

where $S$ is the number of pipeline stages, $B$ is the number of micro-batches (chunks) in a single iteration, $P$ is the number of workers used in the pipeline, $T_F$ is the time cost for a complete forward pass (all forward stages added together) divided by P, $T_B$ is the time cost for a complete backward pass (all backward stages added together) divided by P, and $T_C$ is the communication time for moving a from a worker to its neighbor for the un-overlappable portion of communication.

The bubble ratio is derived from the un-overlappable portions of communication $T_C$ and forward pass $T_B$ (numerator of Equation 1) from the entire end-to-end span of the pipeline (denominator of Equation 1), where $\left(\frac{P^2}{2}-P\right)$ is the gaps/stalls in the pipeline due to lack of components to overlap after the forward and backward passes of the two duplicate models have been overlapped.

Figure 1 illustrates how the bubbles attributed to the dynamic sparsification adds up to, and is different from, the inherent bubbles that are observed in the pipeline scheme.

## B  Pruning, Load Balancing, and Packing

### B.1  Neural Networks Pruning

There are three main considerations that need to be taken into account when applying network pruning: criterion, structure, and schedule of the pruning.

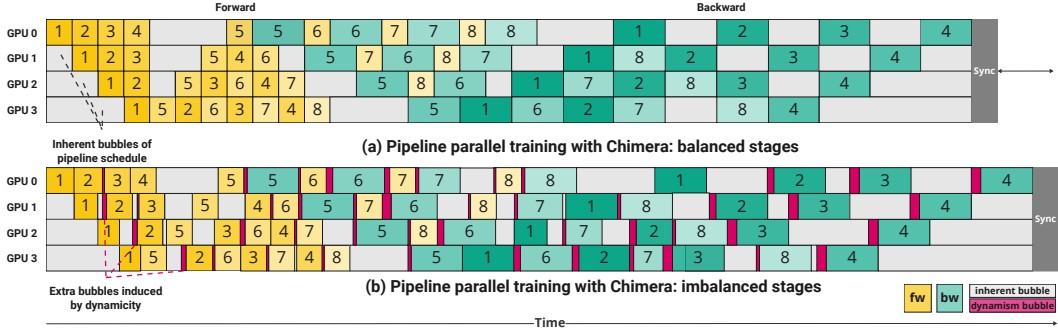

Figure 1: Illustration of bubble types in the Chimera Li & Hoefler (2021) pipeline scheme with 8 microbatches (only one of the duplicated models in Chimera is shown for simplicity). Each row represent a GPU's pipeline stages over time. Inherent bubbles in the pipeline are shown in gray and bubbles introduced by dynamicity (e.g. sparsity) are shown in red.

**Pruning Criterion**: Every pruning scheme needs to define a criterion to choose which parameters to prune. A non-exhaustive list of pruning criteria used in the literature includes: weight magnitude Li et al. (2016); Renda et al. (2020), gradient magnitude Cun et al. (1990); Mozer & Smolensky (1989), Bayesian statistics-based criteria Dai et al. (2018); Molchanov et al. (2017), and reinforcement learning based criteria Lin et al. (2017); He et al. (2018). These criteria can be applied either locally (i.e. considering each layer's weights separately) or globally (i.e. considering weights in all layers).

**Pruning Structure**: Parameters in a model can be removed in a structured or unstructured way. Structured sparsity Kruschke & Movellan (1991) enforces a pattern to be applied while choosing the parameters to be pruned. This can range from removing filters in a convolution layer to removing attention heads in a multi-headed attention layer. On the other hand, unstructured sparsity Han et al. (2015) is not under the constraint of a pattern (i.e parameters can be freely removed), hence, offers a finer granularity. Even though unstructured sparsity offers better flexibility, structured sparsity is more prevalent since it is difficult to implement efficient kernels for sparse data structures in unstructured sparsity and deep learning frameworks have limited support for sparse computations. However, it has been shown that the enforcement of a certain structure for the pruning of parameters can result in significant degradation in model quality compared to unstructured sparsity Kalchbrenner et al. (2018); Elsen et al. (2020).

**Pruning Schedule**: After choosing the criterion and the structure of the pruning, one must decide when to prune and how often to prune. The most popular schedule in the literature consists of pruning after training is over, and then fine-tune the model to recover the loss introduced by the pruning Han et al. (2015). Another effective approach is to remove a certain percentage of weights progressively during the training until the target sparsity is reached Zhu & Gupta (2017), which eliminates the fine-tuning process. There are also schedules that enforce a constant rate of sparsity throughout the training Mocanu et al. (2018).

For a more comprehensive analysis of various sparsification procedures which are applied in deep learning, we refer the reader to Hoefler et al. (2021).

### B.2   GRADUAL GLOBAL MAGNITUDE PRUNING

For our pruning design, we use the gradual pruning schedule proposed in Zhu & Gupta (2017) which is formulated as:

$$S_t = S_f + (S_i - S_f)(1 - \frac{t - t_0}{n\Delta t})^3, \quad t \in \{t_0, t_0 + \Delta t, ..., t + n\Delta t\} \tag{2}$$

where $S_i$, $S_f$, $n$, $t_0$, and $\Delta t$ are initial sparsity, final sparsity, number of pruning steps, initial pruning step, and pruning frequency, respectively. The aim of this schedule is to prune the model rapidly in the initial pruning steps since there are many irrelevant connections, then reduce the pruning rate as the number of parameters in the network gets smaller.

---

**Algorithm 1** Global Pruning Algorithm

---

    **Input:** model, sparsity, rank
    **Output:** model
1: params ← concat_params(model)
2: k ← num_params × (1 - sparsity)
3: local_topk, local_topk_indices ← topk(abs(params), k)
4: topk_values ← gather(local_topk)
5: **if** rank == 0 **then**
6:     global_topk_indices ← topk(abs(topk_values), k)
7: **end if**
8: indices_to_keep ← scatter(global_topk_indices)
9: model = compress_model(model, indices_to_keep)
10: **return** model

---

We employed an unstructured magnitude pruning technique as opposed to a structured one since unstructured magnitude pruning typically retains better accuracy under high sparsity rates Prasanna et al. (2020). Unstructured magnitude pruning is applied globally (taking all parameters in the model

---

**Algorithm 2** Diffusion-based Load Balancing Algorithm

---

    **Input:** loads, num_ranks, max_iters, times, mem_info
    **Output:** transfers (list)
1: transfers ← []
2: **for** iter ← 0 **to** max_iters **do**
3:     total_loads ← [sum($t$) for $t$ in times]
4:     avg_load ← average(total_loads)
5:     var ← variance(total_loads)
6:     status ← ["Overloaded" if $l >$ avg_load
7:             else "Underloaded" for $l \in$ loads]
8:     **for** src ← 0 **to** num_ranks **do**
9:       **if** status[src] == "Overloaded" **then**
10:         dst ← get_least_loaded_rank(loads)
11:         lyr_idx ← get_least_loaded_layer(src, times)
12:         new_loads ← update_loads(src, dst, lyr_idx, loads)
13:         new_total_loads ← [sum($l$) for $l \in$ new_loads]
14:         new_var ← variance(new_total_loads)
15:         mem_req = sum(mem_info[dst]) +
16:             mem_info[src][lyr_idx]
17:         **if** new_var $<$ var && mem_req ¡ MAX_MEM **then**
18:           var ← new_var
19:           loads ← new_loads
20:           update_mem_info(src, dst, lyr_idx, mem_info)
21:           transfers.append((src, dst, lyr_idx))
22:         **end if**
23:       **end if**
24:     **end for**
25: **end for**
26: **return** transfers

---

into account) instead of locally since it has been empirically shown that global pruning yields better accuracy under high compression ratios Blalock et al. (2020).

To our knowledge, there is no deep learning framework that supports global pruning on a distributed model at the time of this writing (support is only for undistributed models). Thus we implemented our own global pruning algorithm as shown in Algorithm 1. The global pruning method takes three arguments, namely the model, target sparsity, and the rank of the device. Note that each rank[1] has only its own portion of the model. First, each rank finds its own local top-$k$ parameters in terms of magnitude (line 3). Then, rank 0 gathers the top-$k$ parameters of each rank (line 4). When rank 0 receives all top-$k$ parameters, it calculates the indices of global top-$k$ parameters to keep (line 6), and sends the indices that belong to each rank (line 8). Finally, after each rank receives its indices to keep, they prune (discard) parameters with all other indices in their local parameters (line 9).

### B.3 LOAD BALANCING

DYNPIPE implements two load balancing algorithms, and can be extended to support other algorithms. The first one is a centralized parameter-based partitioning method that balances partitions based on the number of parameters. We also implemented a version where the same algorithm is used for balancing partitions based on the layer execution times instead of the number of parameters. This algorithm with two variants is built on top of DeepSpeed's load balancing utility functions for partitioning in model parallelism Smith (2023). The second algorithm is an iterative decentralized diffusion-based algorithm that aims to minimize the variance between the workload of each rank by attempting to move layers from overloaded ranks to underloaded ranks in an iterative way. The workload can either be the layer execution times or the parameter counts as in the DeepSpeed-based algorithms. The number of iterations to decide on the final load distribution is a user-defined parameter.

---

[1]We use one MPI rank per GPU.

Algorithm 2 shows the pseudo-code for the diffusion-based load balancing algorithm. After rank 0 gathers the loads (i.e. layer execution times or the number of parameters for each layer) from all ranks, it discovers all layer transfers between ranks by calling a diffusion re-balance function. The number of iterations to minimize the variance is an argument that can be tuned according to the workload. For each iteration of balancing, the total load of each rank, variance, and average load are calculated (lines 3-5). Then, each rank is assigned a status: *overloaded* or *underloaded* (lines 6-7). After the status of each rank is assigned, each overloaded rank attempts to send its least loaded layer to the least loaded rank (lines 7-24). Every time an overloaded rank attempts to send a layer to an underloaded rank, new loads and variance are calculated (lines 12-14). If the new variance is smaller than the current variance and it satisfies the memory constraints of the destination rank, the transfer is accepted and added to the transfers list in the format of (source, destination, layer id) (lines 17-22). When rank 0 discovers all layer transfers from source ranks to destination ranks, it distributes the information to other ranks and the sparse format data structures, CSR, of the layers to be transferred are sent to their new destinations.

We now demonstrate that the two load balancing schemes (used in Algorithm 2) meet the goals for optimal load balancing by using the following lemmas. Detailed proofs on the lemmas are presented in the supplementary material.

**Lemma 1** *A centralized load balancer $L_c$ over $N$ workers satisfies maximum reduction in the imbalance $N_i$ if and only if $N_i$ reduces the bubble ratio to minimum.*

**Lemma 2** *An iterative decentralized diffusion based load balancer $L_d$ over $N$ workers satisfies maximum reduction in the imbalance $N_i$ if and only if $N_i$ reduces the bubble ratio to minimum. Also the load balancer is guaranteed to converge to the maximum reduction in imbalance in the following number of rounds*

$$O\left(\min\left\{N^2\log\left(\frac{SN}{\gamma}\right)\log N, \frac{SN\log N}{\gamma}\right\}\right)$$

*where $\gamma \in \mathbb{R}_{>0}$ is the convergence factor and $S \in \mathbb{R}_{>0}$ is the total number of stages in the pipeline.*

### B.3.1 PROOF OF LEMMA 1

**Lemma 1.** *A centralized load balancer $L_c$ over $N$ workers satisfies maximum reduction in the imbalance $N_i$ if and only if $N_i$ reduces the bubble ratio to minimum.*

*Proof.* We will prove by contradiction. Suppose a centralized load balancer $L_c$ over $N$ workers satisfies maximum reduction in the imbalance $N_i$ when $N_i$ has a bubble ratio higher then the minimum. By the definition of maximum reduction in load balance, $L_c$ must preserve maximum differential between the loads of workers $N_i$ and $N_j$, which $N_i$ and $N_j$ have the minimum load and maximum loads in $N$, respectively. Consequently, increasing the bubble ratio of $N_j$ changes the difference of loads between $N_i$ and $N_j$. This is in contradictory of $L_c$ achieving the maximum reduction on imbalance.

### B.3.2 PROOF OF LEMMA 2

**Lemma 2.** *An iterative decentralized diffusion based load balancer $L_d$ over $N$ workers satisfies maximum reduction in the imbalance $N_i$ if and only if $N_i$ reduces the bubble ratio to minimum. Also the load balancer is guaranteed to converge to the maximum reduction in imbalance in the following number of rounds*

$$O\left(\min\left\{N^2\log\left(\frac{SN}{\gamma}\right)\log N, \frac{SN\log N}{\gamma}\right\}\right)$$

where $\gamma \in \mathbb{R}_{>0}$ is the convergence factor and $\in \mathbb{R}_{>0}$ is the total number of stages in the pipeline.

*Proof.* We leverages core ideas from Lyapunov optimization. We first define a potential function, $\phi$, that measures at each round the total magnitude of workload gaps in the system:

$$\forall r \geq 0 : \phi(r) = \sum_{u,v \in V} |x_u(r) - x_v(r)|$$

Similar to a Lyapunov function, $\phi$ maps the system state (in this case, a vector of workloads for $N$ workers) at any given round to a non-negative scalar value that describes the desirability of the current system state. As $\phi$ decreases toward $0$, the system state becomes more desirable; i.e. the workload is balanced across $N$. As in a standard Lyapunov optimization, we show below that the modifications to a system state caused by executing a single round of our max neighbor algorithm will drift the value of $\phi$ toward zero in a non-decreasing manner. We establish a probabilistic lower bound for the amount of drift in a given round to obtain our time bounds.

For a given round $r \geq 0$ and node pair $u, v \in V$, we define $d_{u,v}(r) = |x_u(r) - x_v(r)|$ to describe the gap between $u$ and $v$'s workload at the end of that round. For each such $r$, we also define: $\{\{u, v\} \mid u$ and $v$ connected and averaged their workloads in round $r\}$, i.e., the set of node pairs that connect and average in $r$, and $D_r = \sum_{u,v \in A_r} d_{u,v}(r - 1)$, i.e., the sum of gaps averaged in $r$. Finally, we define $t_{max}(r) = max_{u,v \in V} \{d_{u,v}(r)\}$ to describe the largest gap between any two nodes at the end of round $r$. From the above analysis that $\phi(r)$ decreases by at least $D_r$ in each round $r$, we proceed to prove the converge time complexity bound.

For a maximum number of rounds to converge to the minimum imbalance:

$$O\left(\min\left\{N^2\log\left(\frac{SN}{\gamma}\right)\log N, \frac{SN\log N}{\gamma}\right\}\right)$$

Note that these two bounds essentially coincide at $\tilde{O}(N^2)$ with $\gamma = \Theta(S/n)$, where the notation $\tilde{O}$ hides logarithmic factors. In other words, if we want all nodes to have the same workload up to a constant factor, the max neighbor strategy uses $\tilde{O}(N^2)$ rounds. We first note that if we arrive at a round r in which $\phi(r) \leq \gamma$, then the system ends this round $\gamma$-converged, i.e. the sum of the gaps is at most $\gamma$, and thus clearly any individual gap is at most $\gamma$. Since $\phi$ is monotonically non-increasing, it follows that every round $r' \geq r$ is also $\gamma$-converged. So we just need to show that with high probability, $\phi$ will decrease to $\gamma$ in the time bound stated by the theorem statement.

For each $r \geq 1$, we call r "good" if and only if $\phi(r-1) - \phi(r) \geq s_{max}(r-1)/(60 \ln(2n))$. We next calculate how many good rounds guarantee that $\phi$ falls below $\gamma$. To do so, we first note that, non-good rounds cannot increase $\phi$, so we are safe to focus only on reductions generated by good rounds in calculating our bound.

By the definition of $\phi$, for each $r \geq 1$ we know that $\phi(r) < s_{max}(r)n^2$. It follows that if $r$ is a good round, then it decreases $\phi(r-1)$ by a multiplicative factor less than $(1 - \frac{1}{60n^2 ln(2n)})$. Finally, we also observe that $s_{max}(0) \leq S$ and therefore $\phi(0) < Sn^2$. Leveraging these observations, to find the number of good rounds needed to decrease $\phi$ below $\gamma$, we just need to find the minimum $s$ time steps such that

$$Sn^2\left(1 - \frac{1}{60n^2 ln(2n)}\right) \leq \gamma$$

A simple calculation implies that $s_{con} = 60n^2 ln(2n) ln(Sn^2\gamma^{-1})$ is sufficient to satisfy this inequality. We have now established that after $s_{con}$ good rounds the system will be $\gamma$-converged for all future rounds. We are left to bound the number of rounds required to generate $s_{con}$ good rounds with high probability.

For each round $r$, let $X_r$ be the random indicator variable that evaluates to $1$ if round $r$ is good and otherwise evaluates to $0$. We know a given round $r$ is good with probability at least $1/N$, regardless of the history of the execution through the round $r - 1$. We cannot, however, directly leverage this observation to calculate (and concentrate) the expected sum of $X$ variables for a given execution length, as the distribution determining a given $X_r$ might depend in part on the outcome of previous experiments. To overcome this issue, we define for each round $r$, a trivial random indicator variable $\hat{X}_r$ that evaluates to $1$ with independent probability $1/N$ and otherwise evaluates to $0$. Note that for each $r$, $X_r$ stochastically dominates $\hat{X}_r$, and therefore for each $s > 0, Y_s = \sum_{r=1}^s X_r$ stochastically

---

**Algorithm 3** Re-pack Layers into Fewer Workers

---

    **Input:** active_gpus, mem_usage
    **Input:** target_num_gpus, num_layers
    **Output:** transfers (list)
 1: transfers ← []
 2: **for** src in range(num_ranks) **do**
 3:     **for** dst in range(src + 1, num_ranks) **do**
 4:         **if** mem_usage[src] + mem_usage[dst] ¡ MAX_MEM
 5:       && sum(active_gpus) ¿ target_num_gpus **then**
 6:           active_gpus[src] = 0
 7:           **for** lyr_idx in range(num_layers[src]) **do**
 8:              transfers.append((src, dst, lyr_idx))
 9:           **end for**
10:           mem_usage[dst] += mem_usage[src]
11:           num_layers[dst] += num_layers[src]
12:         **end if**
13:     **end for**
14: **end for**
15: **return** transfers

---

dominates $s > 0$, $\hat{Y}_s = \sum_{r=1}^{s} \hat{X}_r$. It follows for any $s > 0$, if $\hat{Y}_s \geq s_{con}$ with some probability $p$ then $Y_s \geq s_{con}$ with probability at least $p$.

A Chernoff bound applied to $\hat{Y}_s$, for $s = c.s_{con}$ (where $c \geq 1$ is a sufficiently large constant defined with respect to the constants in $s_{con}$ and the constants in the Chernoff form used), provides that $\hat{Y}_s \geq s_{con}$ with high probability, and therefore so is $Y_s$. To conclude the proof, we note that $c.s_{con} \in O\left(N^2 log(\frac{SN}{\gamma})logN\right)$, as required by the theorem $\gamma$ statement.

### B.4 RE-PACKING DYNAMIC MODELS TO FEWER WORKERS

Workload re-packing is the process of merging the total workload into a smaller number of worker (GPUs) with the purpose of using the available resources more efficiently, i.e. unused resources can be released. This can be achieved with simple algorithms (in small scale) such as first-fit, best-fit, and round-robin as well as complex optimization problems (for large scale) such as ant colony optimization Dorigo et al. (2006) or genetic algorithms Dasgupta et al. (2013). Workload packing aims to increase GPU utilization and reduce the overall number of GPUs employed to continue the training process. For long training schedules that are common in LLM training, workload packing can result in substantial cost savings. It may also provide improved performance due to reduction in the number of cross-GPU communication calls, and smaller pipeline bubbles.

Algorithm 3 shows a first-fit algorithm that we used for workload consolidation. We iterate over all the available GPUs (lines 2-3) and check if the combined memory usage of the two GPUs is less than the maximum memory capacity of a single GPU, and the number of active GPUs is greater than the target number of GPUs *target_num_gpus* for packing (lines 4-5). If that is the case, we transfer all layers of the source GPU to the destination GPU (lines 7-8). Then, it updates the memory usage and the number of layers on the destination GPU accordingly. This process continues until all the available GPUs have been checked and processed. The goal of this algorithm is to reduce the number of active GPUs to the *target_num_gpus*, while also ensuring that the total memory usage remains within device limits.

## C IMPLEMENTATION

The DYNPIPE load balancing system was developed on top of Megatron-LM v3.0 [2]. Each component of DYNPIPE, namely pruning, load balancing, and re-packing is implemented in a separate package for ease of use and extension.

---

[2]https://github.com/NVIDIA/Megatron-LM/releases/tag/v3.0.2

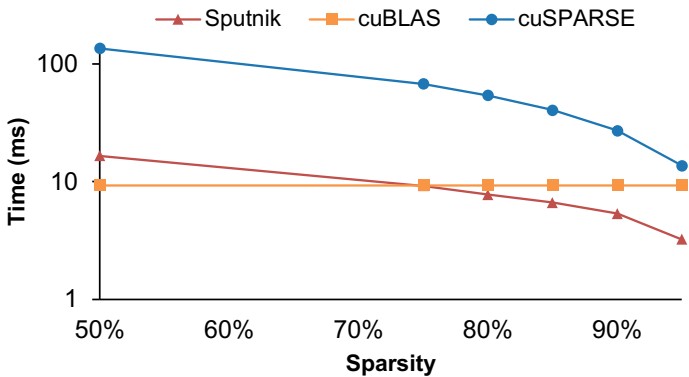

Figure 2: Sparse (Sputnik Gale et al. (2020) and cuSPARSE) vs Dense (cuBLAS) matrix multiplication performance comparison for M=N=K=4096 on Nvidia A100. Starting at 75% sparsity level, sparse kernels using Sputnik gives performance advantages over dense kernels.

Unstructured pruning requires a sparse storage format to compactly store, train, and transfer the pruned model. One of the most commonly used sparse formats is the compressed sparse row (CSR) format. Using a sparse matrix format requires dense matrix multiplication (DMM) operations to be converted to sparse counterparts (SpMM). Since PyTorch does not support computing the derivative of SpMM operations for backpropagation on a CSR tensor, we evaluated CSR-based SpMM implementations available for use on GPUs, namely cuSPARSE by Nvidia and Sputnik Gale et al. (2020). Figure 2 shows the performance of cuSPARSE and Sputnik against the dense counterpart (cuBLAS). The SpMM kernel of Sputnik outperforms cuSPARSE in all sparsity levels. This is mainly because Sputnik kernels were implemented by specifically considering the deep learning workloads, unlike cuSPARSE kernels that mainly target the HPC workloads, which often have more than 99% sparsity. It is also worth noticing that Sputnik starts to outperform cuBLAS after 75% sparsity. Thus, for sparse operations, we implemented PyTorch bindings for the CUDA kernels of Sputnik [3].

The gather and scatter operations in global pruning were implemented by employing NCCL Peer-to-Peer (P2P) send-receive operations instead of collective communication operations since the sizes of the objects to be sent (local_topk) and received (indices_to_keep) from each rank are different and other ranks do not have this size information to participate in the collective call.

The necessary information for load balancers such as layer execution times and memory usage comes from the profiling iteration after each pruning iteration. The execution time profiling is implemented by extending the built-in timers of Megatron-LM. The memory consumption of each pipeline stage is gathered with PyTorch's memory statistics for CUDA.

## D  ADDITIONAL RESULTS AND ABLATION

### D.1  OVERHEAD OF LOAD BALANCING

The time spent to load balance the model is negligible in deep neural networks since they are typically trained for days if not months Chowdhery et al. (2022); Hoffmann et al. (2022). Table 1 shows the time spent while load balancing for different balancers in terms of the number of iterations. The maximum number of iterations for the diffusion algorithm is set to 5 but our experiments showed that it usually converges after two iterations. The reported load balancing times include both the load balancing decision and the actual transfer of the parameters and index data structures (i.e. row offsets and column indices in CSR format) of the layers to be sent or received. Among all balancers, *Diffusion by Time* has the least overhead. Considering the fact that the frequency of pruning is in the order of 1000s-10000s to recover the accuracy after pruning Gale et al. (2019); Zhu & Gupta (2017), the load balancing overhead is easily amortized. All our throughput and speedup results include the load balancing overhead unless specified otherwise.

---

[3]The Sputnik bindings are made available at the following link: `https://anonymous.4open. science/r/Torch-Sputnik-E926/README.md`.

Table 1: Load balancing overhead in terms of number of training iterations. Since models train to 10,000 iterations, the overhead is effectively negligible.

| # of Layers | Partition by Time | Partition by Param | Diffusion by Time | Diffusion by Param |
|---|---|---|---|---|
| 24 | 25 | 61 | 12 | 18 |
| 32 | 9 | 55 | 7 | 20 |
| 40 | 12 | 56 | 11 | 18 |
| 48 | 13 | 54 | 4 | 13 |
| 56 | 14 | 59 | 9 | 13 |

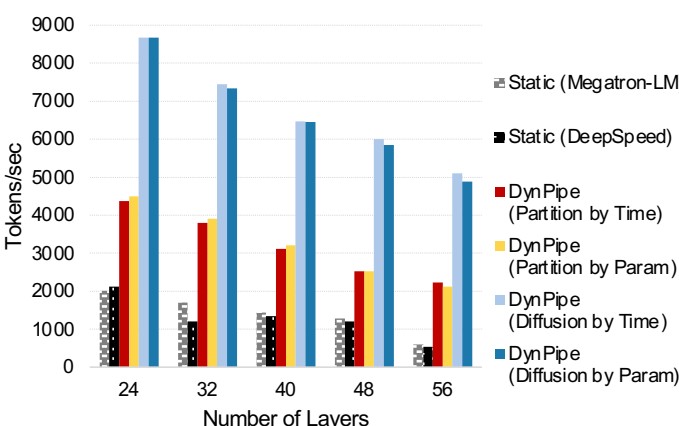

Figure 3: Multi-node end-to-end training throughput of GPT models for a network with unstable bandwidth. We inject network delays up to 4x modeled by a normal distribution (as suggested by Sukhov et al. Sukhov & Kuznetsova (2009)). The intended delay is achieved by inflating the MPI message sizes based on the delay model.

### D.2 Effect of Change in Network Bandwidth on Load Balancing

In fact DYNPIPE's load balancing algorithm is desgined to, indirectly, handle variability/instability in the network. That is since we define our diffusion load balancing algorithm to consider the gaps between workers to include the total time until work arrives to node B from neighboring node A, i.e. we include the amount of time that A spent on work plus the time it takes to transfer the activations of the layers over the network. In intuitive terms, if worker B is stalling due to delay from neighbor A (in part due to a slow network connection between A and B), the load balancer would push more work to worker B until the amount of work in A plus the time it takes to transfer the activations over the network is roughly equal to the amount of work on B. Figure 3 shows results in a multi-node setting where we inject up to 4x delay in exchange of layers between neighbor nodes to demonstrate the robustness of DYNPIPE load balancing w.r.t. fluctuations in the network bandwidth. In fact the improvement of speedup of DYNPIPE over the baseline static model increases since the static model suffers from higher stalling when the network bandwidth fluctuates due to contention for instance.

### D.3 Vertical Scaling

In single-node multi-GPU vertical scaling experiments, the number of layers in the model and the number of GPUs used in the pipeline are changed. In Table 2, we report throughputs of the static baseline balancer (Megatron-LM) and the best-performing dynamic load balancers from end-to-end training experiments (*Diffusion by Time* and *Partition by Time*). The dynamic load balancers speed up the training in various degrees up to 1.39x for different numbers of GPUs.

Table 2: Vertical scaling experiments show the throughputs (samples/sec) of baseline Megatron-LM, and time-based algorithms, namely *Diffusion by Time* and *Partition by Time* where the target sparsity is 90%. The speed is calculated for the best-performing balancer in each case. The benefits of dynamic load balancing increase as the number of GPUs in the pipeline increases.

| # Layers | # GPUs | Megatron LM | Diff by Time | Part by Time | Speed Up |
|---|---|---|---|---|---|
| **24** | 2 | 10.67 | 12.38 | **12.88** | 1.20x |
| | 4 | 20.23 | **24.86** | 24.82 | 1.22x |
| | 8 | 37.253 | **46.939** | 45.071 | 1.26x |
| **32** | 2 | 8.28 | **10.35** | 10.12 | 1.25x |
| | 4 | 15.69 | 19.06 | **19.76** | 1.26x |
| | 8 | 30.809 | **39.899** | 37.933 | 1.29x |
| **40** | 2 | 7.14 | 8.84 | **9.13** | 1.28x |
| | 4 | 12.84 | 16.11 | **16.82** | 1.31x |
| | 8 | 26.425 | **35.262** | 33.296 | 1.33x |
| **48** | 2 | OOM | OOM | OOM | OOM |
| | 4 | 10.89 | **14.7** | 14.54 | 1.35x |
| | 8 | 23.126 | **31.724** | 29.711 | 1.37x |
| **56** | 2 | OOM | OOM | OOM | OOM |
| | 4 | OOM | OOM | OOM | OOM |
| | 8 | 19.126 | **26.724** | 25.711 | 1.39x |

Figure 4: Re-packing the layers of GPT models into fewer GPUs as the model gets smaller due to gradual pruning. Left Y-axis: throughput/number of GPUs. Right Y-axis: throughput (tokens/sec). **Below**: we show the average number of GPUs needed throughout the training at which we dynamically re-pack (total 10,000 iterations).

One important observation is that as the number of GPUs used in the pipeline increases, the speed-up gained by the usage of a dynamic balancer builds up. This suggests that the importance of load balancing increases as the pipeline gets deeper because the additional bubbles that are introduced by the dynamic nature of the model affect the efficiency of the pipeline more. This is important when considering the fact that the model size of large language models doubles approximately every 3.9 months Zhang et al. (2022) which leads to deeper pipelines.

## D.4  AVERAGE NUMBER OF GPUs USED IN RE-PACKING

Figure 4 reports the throughput/number of GPUs for each model size where the model is packed into 6, 4, and 2 GPUs. The 8 GPU setting for each model size serves as a baseline where there is no packing. At the bottom part of the figure we shows how the average number of GPUs used change in the course of 10,000 training iterations.

## D.5  DYNAMIC MINIBATCH/MICROBATCH SIZE

In cases where the total load of the pipeline decreases such as gradual sparsification and freeze training, carefully changing the minibatch and microbatch size according to the needs of the new pipeline after load balancing may increase the efficiency of the training. For instance, GPipe Huang et al. (2019) suggests the number of micro batches to be greater than four times the number of GPUs in the pipeline for optimal overlapping. Since the packing decreases the number of GPUs in the pipeline, adjusting the number of micro batches in the pipeline after packing could be beneficial. In addition, minibatch size can be increased after the pruning operations since the memory requirement for execution is less after the pruning. DYNPIPE currently does not support this feature, which if supported would further improve the speedup gains.

## E  RELATED WORK

### E.1  LOAD BALANCING MODEL-PARALLEL DEEP NEURAL NETWORKS

#### E.1.1  LAYER-WISE LOAD BALANCING

Layer-wise balancing techniques work on layer granularity instead of operators. DeepSpeed Microsoft (2023) offers three partitioning methods to balance the workload of stages: parameters, uniform, and regex. While the parameters method is trying to balance the number of parameters in each stage, the uniform aims to distribute the layers evenly. Regex only distributes the layers that match the given regex (e.g. transformer layers). Similar to the parameters method of DeepSpeed, He et al. He et al. (2021) balance the stages based on the number of parameters in each stage. Narayanan et al. Narayanan et al. (2021) assign each stage the same number of transformer layers to balance the load. None of the aforementioned studies use the actual execution time of the layers to decide on the distribution of layers. DYNPIPE supports DeepSpeed's partitioning scheme with both parameters and layer execution times to guide load balancing, as well as a diffusion-based load balancing algorithm out of the box.

#### E.1.2  LOAD BALANCING VIA GRAPH PARTITIONING

Graph partitioning-based load balancing schemes find atomic operations in the model and consider them as nodes in a directed acyclic graph (DAG). Edges in the graph represent the dependencies between operations. Tanaka et al. Tanaka et al. (2021) partition the DAG in three phases at which they first find atomic operations, then group these operations into blocks according to their computation times, and finally, they combine blocks into final partitions by using a dynamic programming-based algorithm. Qararyah et al. Qararyah et al. (2021) create disjoint clusters from the nodes of the graph by finding critical paths and mapping these clusters to devices based on a mapping algorithm that takes both critical-communication minimization and temporal load balancing into account. Both studies perform profiling before the actual training and partition the graph once.

#### E.1.3  LOAD BALANCING IN MIXTURE OF EXPERTS MODELS

The mixture of experts (MoE) Jacobs et al. (1991) models contain many sub-networks (experts) where a router allocates inputs to top-k experts. At scale, experts are distributed across devices. Lepikhin et al. Lepikhin et al. (2020) defines an expert's capacity to limit the maximum number of tokens that can be processed by an expert to achieve workload balance. Fedus et al. Fedus et al. (2022) route each token to only one expert and use the same expert capacity for restrictions. Lewis et al. Lewis et al. (2021) employ an auction algorithm Bertsekas (1992) to solve the token-to-expert assignment problem. This line of work is different from ours in the sense that their aim is to balance

workload in the feed-forward network while our work aims to balance all layers of the transformer model.

## E.2 PACKING

In dynamic neural network models, packing the total workload into fewer number accelerators can provide significant cost-saving benefits. PipeTransformer He et al. (2021) offers an elastic pipelining system for freeze training where some of the layers of the model are frozen during the training. PipeTransformer packs the remaining active layers into fewer GPUs and creates pipeline replicas if possible. When PipeTransformer receives a notification for layer freezing, it attempts to divide the number of GPUs by 2 subject to the memory capacity constraints. On the other hand, our work DYNPIPE can pack to an arbitrary number of GPUs specified by the user. Another difference between the packing mechanism of DYNPIPE and PipeTransformer is that PipeTransformer uses the parameter size as a proxy to estimate the memory usage while DYNPIPE uses the actual memory usage from the profiling step before load balancing. Finally, PipeTransformer is only capable of packing layers to fewer GPUs, and not load balancing. DYNPIPE, on top of being capable of re-packing when deemed beneficial, it can also redistribute the workload to achieve a better load balance.

## E.3 DYNAMIC PRUNING

Model pruning is a fast-paced research area. Since the optimization problem has many dimensions, there are many approaches to prune a model. We mainly focus on the schedule of the pruning rather than the decision of how to prune (e.g. magnitude pruning, variational dropout etc.) and what kind of structure (e.g. unstructured pruning, structured pruning) to be applied while pruning.

One of the commonly used sparsification technique is sparsification during training (i.e. gradual pruning) where the pruning starts before the model is trained until convergence. While some studies Wortsman et al. (2019); Lin et al. (2020) use a binary mask to specify whether a parameter is pruned, which enables them to apply better weight regrowth or selection, others Gale et al. (2020) delete the pruned parameters to reduce the memory usage and number of operations. There are also many works on how fast to prune. For instance, Zhu and Gupta Zhu & Gupta (2017) prune the model rapidly in the first pruning steps when there are many abundant parameters in the model, and then reduce the pruning ratio as the number of parameters in the model are getting less and less. Dai et al. Dai et al. (2019) employ a three phase schedule (birth-brain, baby-brain, and adult-brain) similar to the human brain development. Mostafa et al. Mostafa & Wang (2019) uses magnitude pruning as criterion to prune the parameters and regrows parameters to comply with the training budget.