# OpenReview forum: "Elastic Load Balancing for Dynamic LLMs"
_ICLR.cc/2024/Conference — Submitted to ICLR 2024_

### Official Review · Reviewer_fgMF · 2023-10-31

**Soundness:** 2 fair
**Presentation:** 1 poor
**Contribution:** 2 fair
**Rating:** 5
**Confidence:** 4

**Summary:**

This paper introduces DynPipe, a system to support LLM training tasks with running time dynamics, where such methods include neural network pruning, layer frozen training, etc. A system is implemented to support the load balancing in such training tasks. The balancing policies include centralized partitioning based on parameters and decentralized partitioning based on workloads. Empirical study was conducted to verify the effectiveness and efficiency of the proposed design.

**Strengths:**

- To build an efficient system to support LLM training tasks with runtime dynamics is interesting research from the system perspective.

- The paper leveraged a production-level cluster for some demonstration of the deployment of the system.

**Weaknesses:**

- The paper is pool written and hard to understand:

  - The paper makes some exaggerated statements about its contribution. For example, "Research on dynamic models will not deliver practical impact unless there is a platform from which those models can be made efficient." -- This is untrue from the machine learning algorithm's perspective. As long as such an algorithm shows statistical efficiency or better generalization performance, it has a significant practical impact w/wo a platform/framework, right?

  - The technique session is confusing and not self-explained; for example, in Lemma 1 and 2, the term "bubble" is referred to without a formal definition.

  - The experimental section is poorly organized; there is even a lack of formalization of the central hypothesis of the evaluation.

- There is also some issue w.r.t the baseline selection, Megatron is designed to support standard non-sparse LLM training, DeepSpeed is similar where the additional effort is made for MOE. Those are not strong baselines for such training tasks. As far as I know, systems like PEFT from huggingface include some relevant functionality.  Some more advanced baselines should be considered for evaluation.

**Questions:**

Would it be possible to provide an empirical study with state-of-the-art baselines?

---

> ### Author Response · Authors · 2023-11-11
>
> >The paper makes some exaggerated statements about its contribution. For example, "Research on dynamic models will not deliver practical impact unless there is a platform from which those models can be made efficient." -- This is untrue from the machine learning algorithm's perspective. As long as such an algorithm shows statistical efficiency or better generalization performance, it has a significant practical impact w/wo a platform/framework, right?
>
> We argue that it if there is no effective high-throughput solutions (like the one in this paper) that allow for researching the throughput effectiveness of methods like pruning/freezing techniques at scale (and we emphasize "at scale"), researchers would and are limited in studying those methods on a small scale, which is a problem. For instance, without an effective methods for studying dynamic GPT3-sized models on up to 1000 GPUs (as in this paper), the true potential of dynamic models on large models can not be realized. That being said, we can tune down this claim in the paper in the revised PDF to not give wrong impressions to the reader.
>
> >The technique session is confusing and not self-explained; for example, in Lemma 1 and 2, the term "bubble" is referred to without a formal definition.
>
> Bubbles are formally defined in Supp. material Section A.
>
> >The experimental section is poorly organized; there is even a lack of formalization of the central hypothesis of the evaluation.
>
> We will revise the structure of that section in the revised PDF, and also add a summary of the key findings.
>
> >There is also some issue w.r.t the baseline selection, Megatron is designed to support standard non-sparse LLM training, DeepSpeed is similar where the additional effort is made for MOE. Those are not strong baselines for such training tasks. As far as I know, systems like PEFT from huggingface include some relevant functionality. Some more advanced baselines should be considered for evaluation.
>
> PEFT, and other baselines, are not applying true sparse computation (i.e. they just mask the pruned values to zero and use dense MatMult), and hence the real imbalance that comes from using sparse models would not manifest. Hence comparing to PEFT would be akin to comparing to a dense computational baseline (similar to the case of Megatron and DeepSpeed). Section C in the supp. material describes how we had to implement bindings for really sparse operations.

---

> > ### Comment · Reviewer_fgMF · 2023-12-04
> > **Thank you for your feedback**
> >
> > Thank you for your feedback! I adjusted my score.

---

### Official Review · Reviewer_cgYk · 2023-11-01

**Soundness:** 2 fair
**Presentation:** 2 fair
**Contribution:** 2 fair
**Rating:** 3
**Confidence:** 5

**Summary:**

DYNPIPE is introduced as a tool that enables the exploration of dynamic models, substantially enhancing the end-to-end training efficiency and making their practical application more viable. It operates independently of the underlying pipeline parallelism, pruning, and freezing schemes, ensuring compatibility with various compute optimization and reduction strategies. To mitigate the adverse effects of dynamic models on pipeline utilization, load balancing algorithms are proposed, which are proven to converge to optimal balancing. The framework's benefits are showcased through gradual pruning training and layer freezing scenarios across both single-node and multi-node settings, including a strategy to reduce GPU usage by consolidating work onto fewer GPUs. DYNPIPE demonstrates significant speed improvements over existing solutions, achieving up to 1.3x speedup on a single-node with 8 A100 GPUs, over 2.5x speedup in multi-node settings with up to 720 A100 GPUs, and an average of 2.3x speedup over the state-of-the-art layer freezing solution, all while effectively reducing GPU requirements by half without compromising performance.

**Strengths:**

Load balancing algorithms for adaptive equalization of compute workloads among different workers

**Weaknesses:**

Unclear presentation of the solution strategy and how the problem and solution is different from prior works.
Incomplete analysis because the communication aspects are not discussed.

**Questions:**

1) In Figure 1, how is the Imbalanced pipelines in dynamic models lead to additional stalling in data parallelism showed in the figure. It seems like the idleness is not as high as pipeline in each dense level.
2) In Figure 2, what is P2P layer transfer? As far as pipeline is concerned, what is GPU1 doing when GPU 0 is occupied. For example, in stage, L5 send data to L4 after finishing L1, L2, L3, L5. What is GPU 1 doing? Is it idle?
3) How is the prediction result influenced by the pruning?
4) During the measurement of the throughput, how to reflect that profiling time and other pruning time will not influence the total performance of the framework?
5) Why y-axis in Figure 4 and 5 use token/# GPU. Is it same as token per device? Meanwhile, is the communication bandwidth between GPU sufficient to not influence the experiment?
6) In figure 4, why is the throughput of the model with different number of GPUs roughly the same. Shouldn’t we expecting the throughput will increase accordingly?
7) In section 4.1, how is the prune region and iteration being selected. What is the affect of changing the settings?
8) In the description of Diffusion by Param and Diffusion by Time, how do they iteratively minimize load variances among accelerators? Do they communicate or they select parameters based on the variance of after gradient decent?
9) In figure 4, why only partition by param is included?
10) In figure 5(right), it seems that there is not a clear relationship between number of GPU and throughput/# GPUs. Why is that? The goal is the increase the throughput after pruning in order to use less computing resources. Moreover, if we multiply # of GPU with throughput/# GPUs, the total throughput of the system will decrease. Why is that?

---

> ### Author Response · Authors · 2023-11-11
>
> >- In section 3.2, what is callable primitive operator stans for in op_function, invars, outvars, and spmd_rules?
>
> >-In figure 2, what is the MetaIR? There is a MetaIR on the left side and another MetaIR in the middle.
>
> >-In Section 3.3 Sharedim(id=j)(Sj) is used to determine the parallelism strategies. How many such strategy could be applied? I notice the paper mentioned S1 S2 S3 in the rest of the paper. For Si, how many ways to partition the tensor?
>
> >-What is the definition of score function? Is it just the train time? If two parallelisms could be applied, how to evaluate which one is better?
>
> I think there is some mistake here. Those questions are not about this paper (citing things that do not exist). Perhaps they were copied by mistake from a review intended to another paper?
>
> >-How is the communication bandwidth affect the result?
>
> Supp. material (Section D.2 "Effect of Change in Network Bandwidth on Load Balancing") show an ablation study on the effect of network bandwidth (Figure 3)
>
> >In figure 4, why is the throughput of the model with different number of GPUs roughly the same. Shouldn’t we expecting the throughput will increase accordingly?
>
> The results in that figure are for weak scaling (as shown on the label above the figure). The amount of work increases with the number of GPUs, hence no performance improvement is to be expected.
>
> >In section 4.1, how is the prune region and iteration being selected. What is the affect of changing the settings?
>
> They follow the pruning/freezing strategy; load re-balancing is done every time the model changes.
>
> >In the description of Diffusion by Param and Diffusion by Time, how do they iteratively minimize load variances among accelerators? Do they communicate or they select parameters based on the variance of after gradient decent?
>
> Diffusion happens in a decentralized way where each GPU passes some of its load to the neighbor if it has more work that its neighbor. This is repeated iteratively until each GPU has the same amount of work. This decentralized approach is particularly beneficial when scaling to a large number of GPUs, since no collective communication between GPUs would be required.
>
> >In figure 4, why only partition by param is included?
>
> In that figure we report "partition by param" only since it is the highest performing baseline.
>
> > However, the related work discussion is missing. It would be good to discuss prior work and the new challenges faced in dynamic DNNs.
>
> The related work section in the supp. material (Section E) covers load balancing approaches in computing systems. That being said, we agree it should be expanded to give a wider give (which what we will do in the revised PDF).
>
> >In figure 5(right), it seems that there is not a clear relationship between number of GPU and throughput/# GPUs. Why is that? The goal is the increase the throughput after pruning in order to use less computing resources. Moreover, if we multiply # of GPU with throughput/# GPUs, the total throughput of the system will decrease. Why is that?
>
> There is a relation: Throughput/ #GPUs tends to go up as the number of GPUs decrease; that is since the more the GPUs, they more pronounced the effect of load imbalance will be, and hence the throughput/ #GPUs will drop. That is akin to how throuhput/GPU is higher in single GPU run than multi-GPU runs since you avoid the communication cost.On the other hand, the aggregate throughput (right y-axis) will drop if you use fewer GPUs.

---

### Official Review · Reviewer_4Zma · 2023-11-10

**Soundness:** 2 fair
**Presentation:** 2 fair
**Contribution:** 1 poor
**Rating:** 3
**Confidence:** 4

**Summary:**

Given the significant computational and memory costs involved in training Large Language Models (LLMs), recent studies have proposed various techniques to curtail these expenses. These techniques include dynamically pruning or freezing portions of the model. However, these methods can lead to an imbalance in the workloads within pipeline parallelism, as some workers will have fewer tasks after the pruning or freezing processes. To address this issue, the paper proposes a solution that dynamically rebalances workloads among distributed training workers. The results indicate that this approach surpasses static load balancing baselines in achieving higher training throughput.

**Strengths:**

1. **Addressing a Timely Issue**: The paper addresses a timely problem associated with combining pipeline parallelism with dynamic pruning and freezing. The clear motivation behind this problem is well-illustrated in Figure 1.

2. **Enhancing Reproducibility**: The authors have contributed to reproducibility by providing a source code repository. This repository not only allows for easy reproduction of the results but also enables other users to utilize the system to improve training time using dynamic schemes.

3. **Extensive Evaluation**: The paper undertakes a large-scale evaluation with a substantial GPU cluster comprising 720 A100 GPUs. This evaluation substantiates the scalability of the proposed solution.

**Weaknesses:**

1. **Limited Novelty**: The paper's technical contribution is limited due to its use of existing techniques in a new context, rather than introducing entirely new concepts. The proposed load balancing solutions seem to involve applications of DeepSpeed's workload partitioning algorithms at the end of each pruning, and the diffusion-based algorithm also employs a similar partitioning strategy, albeit more akin to work stealing.

2. **Lack of Overhead Discussion**: The paper does not adequately discuss the overhead of different strategies, which could have provided an interesting perspective on these solutions. It remains unclear whether the throughput reported in the evaluation takes into account all overheads.

3. **Absence of Key Training Metrics in Evaluation**: The evaluation does not include important training metrics such as time-to-accuracy or learning curves. Even though model accuracy is not the primary focus of the proposed solution, including this information would have added to the completeness of the paper. For instance, it might be the case that improvements in throughput during large-scale training do not translate into time-to-accuracy gains because the pruning/freezing techniques reduce training quality. This potential issue could limit the usefulness of such solutions.

**Minor**: The paper does not adhere to the proper citation format (it should use \citep instead of \citet). This oversight can hinder readability and understanding at certain points in the paper.

**Questions:**

1. Could the authors provide a detailed discussion on the overheads associated with the different strategies and clarify whether these overheads were factored into the reported throughput in the evaluation?

2. Would it be possible for the authors to include key training metrics, such as time-to-accuracy and learning curves, in the evaluation? This inclusion could provide a more comprehensive understanding of the impact of the proposed solutions on training quality.

---

> ### Author Response · Authors · 2023-11-11
>
> >Limited Novelty: The proposed load balancing solutions seem to involve applications of DeepSpeed's workload partitioning
>
> algorithms at the end of each pruning
> To the authors knowledge, this paper is the first to demonstrate provably practical solutions, at scale, for load balancing dynamic models.  We use DeepSpeed API to just move the parameters between GPUs; DeepSpeed itself had no workload partitioning algorithms that we build upon.
>
> >Lack of Overhead Discussion
>
> The supplementary material (Section D.1 "Overhead of Load Balancing") has a detailed analysis of the overhead (Table 1 in Supp. material)
>
> >Absence of Key Training Metrics in Evaluation:  it might be the case that improvements in throughput during large-scale training do not translate into time-to-accuracy gains because the pruning/freezing techniques reduce training quality
>
> We argue that it is the other way around: if there is no effective high-throughput solutions (like the one in this paper) that allow for researching the throughput effectiveness of methods like pruning/freezing techniques at scale (and we emphasize "at scale"), researchers would and are limited in studying those methods on a small scale, which is a problem. For instance, without an effective methods for studying dynamic GPT3-sized models on up to 1000 GPUs (as in this paper), the true potential of dynamic models on large models can not be realized.
>
> >Would it be possible for the authors to include key training metrics, such as time-to-accuracy and learning curves, in the evaluation?
>
> We will include those numbers to the revised PDF. Note that we will not be able to include results for the largest sized model since training those models to completion can take weeks (and the deadline is in 11 days).

---

### Author Response · Authors · 2023-11-11

We thank the reviewers for the comments and feedback.

We like to point out that the primary goal and impact of the paper is not just to make the dynamically pruned LLMs faster. Though we emphasize that we do achieve for GPT models w/ up to 56 layers

- 1.4x speedup over Megatron-LM and DeepSpeed static balancing on single node with 8 A100 GPUs, and
- 5.6x speedup on 720 A100 GPUs for data+pipeline parallelism for GPT

The primary aim and impact, we argue, is that research on dynamic models will not deliver practical impact unless there is a platform from which those models can be made efficient. For instance, the new Figure 1 is showing that we are at 3x the idleness in pruned models over dense models if we want to practically run dynamic GPT models in a data+pipeline setting, and that obviously inhibits any efforts toward realizing practical efficiency for dynamic schemes such as pruning or MoEs.

With DynPipe, we are providing a solution that allows researchers to explore dynamic models, since any pipeline scheme, pruning scheme, or load balancing scheme can be plugged to DynPipe. To that end we also implemented global pruning on distributed models on PyTorch, PyTorch bindings to SoTA library of SpMM operations, and most importantly load balancers that are proven to converge.

---

### Meta-Review · Area_Chair_SS9X · 2023-12-19

**Metareview:**

The reviewers have concerns about novelty (and how this paper differs and improves over previous work), and evaluation. The general feeling is that these need to be improved before this paper can be accepted.

**Justification For Why Not Higher Score:**

Concerns about novelty (and how this paper differs and improves over previous work), and evaluation.

**Justification For Why Not Lower Score:**

N/A

---

### Decision · Program_Chairs · 2024-01-16

Reject